# Genomics and metabolomics assisted functional characterization of *Bacillus velezensis* D83 as a biocontrol and plant growth-promoting bacterium

Jili Chen,[1,2] Jiabang Dong,[1,2] Zhiwen Xiao,[3] Dong Jiang,[4] Jialong Yu,[4] Lương Hùng Tiến,[5] Hoàng Trung Tín,[5] Xiaojun Zhao,[4] Tao Liu[1,2]

**ABSTRACT** Diseases triggered by phytopathogenic bacteria lead to substantial economic losses within the realm of agricultural production. *Bacillus velezensis* is an important agricultural biocontrol and plant growth-promoting bacterium. In this study, we evaluated the inhibitory ability of *B. velezensis* D83 against eight plant pathogenic fungi. D83 has a broad spectrum of antagonistic activity against these eight plant pathogens. In addition, our study found that D83 promotes seed germination and plant growth in maize and tobacco, and this growth-promoting ability is related to its growth-promoting properties such as indoleacetic acid production, nitrogen fixation, and phosphate solubilization. This effect of promoting plant growth is more effective than the microbial preparations on the market. A complete genome sequencing of D83 yielded a genome size of 3,929,757 bp and a GC content of 46.5%. The genome analysis identified 3,987 protein-coding genes. Gene clusters for the antimicrobial metabolites fengycin, surfactin , bacilysin, butirosin A/B, bacillaene, difficidin, macrolactin, surfactin, and bacilysin were identified in D83. Genes associated with plant growth promotion were also identified and analyzed. Analysis of the amplicons of genes associated with fengycin, surfactin, and iturin synthesis revealed that D83 has the capacity to encode the genes responsible for fengycin, surfactin, and iturin synthesis. In addition, liquid chromatography-tandem mass spectrometry (LC-MS/MS) analysis of D83 cell sediment extracts confirmed the presence of antifungal metabolites and plant growth-promoting hormones. These results suggested that D83 has the combined ability to promote plant growth and antagonize common plant pathogens and is a source for further development of antimicrobial compounds.

**IMPORTANCE** D83 has a broad-spectrum inhibitory effect on eight plant pathogenic fungi. D83 also demonstrated the properties of a variety of plant-promoting bacteria. Under *in vitro* conditions, D83 has the ability to promote the germination of maize and tobacco seeds, and in pot experiments, D83 has the ability to promote the growth of maize and tobacco seedlings and suggests that promotion is more pronounced than that of sterile water treatment (CK) and commercially available microbial preparations treatment (A). The gene cluster responsible for the production of the antimicrobial metabolites fengycin, surfactin, bacilysin, butirosin A/B, bacillaene, difficidin, macrolactin, surfactin, and bacilysin was identified in the D83 genome by genome prediction analysis. The genes responsible for fengycin, iturin, and surfactin biosynthesis were identified by D83 amplicon analysis. Furthermore, LC/MS analysis of D83 fermented extracts confirmed the presence of antifungal metabolites and plant growth substances.

**KEYWORDS** *Bacillus velezensis*, D83, antimicrobial activity, genome analysis, plant growth promotion

**Peer Reviewer** Changyi Zhang, University of Illinois Urbana-Champaign Carl R Woese Institute for Genomic Biology, Urbana, Illinois, USA

Address correspondence to Xiaojun Zhao, 626793262@qq.com, Tao Liu, yantao618@126.com, Jiabang Dong, 1913583877@qq.com, Zhiwen Xiao, 2561250282@qq.com, or Jili Chen, 2332710778@qq.com.

Jili Chen, Jiabang Dong, and Zhiwen Xiao contributed equally to this article. Author order was determined by drawing straws.

Z.X. is employed by Yunnan Tobacco Company Yuxi Branch, Yimen Branch, Yimen, China. X.Z. and D.J. are employed by Yunnan Tobacco Company Kunming Branch, Songming Branch, Songming, China. The other authors declare no conflict of interest.

10.1128/spectrum.00300-25 **1**

Plants are susceptible to diseases that cause crop losses in plant production. Although caused by bacteria, fungi, and viruses among microorganisms, fungi are considered to be the most widespread and destructive parasites in plants (1). Although the use of chemical pesticides can be effective in controlling a wide range of diseases of major crops, the overuse of chemicals can have negative impacts on the environment and human health (2). Biological control is often considered a safe method compared to chemical pesticides (3). Biological control includes natural competitors or disease-suppressing microorganisms (biocontrol agents), genes, or metabolites that act in different ways in controlling fungal plant diseases as a means of mitigating pathogen populations and thereby improving plant health (4, 5). Biocontrol agents used in agriculture are bacterial and fungal species, including *Streptomyces*, *Paenibacillus*, *Pseudomonas,* and *Bacillus* genera, which can protect plants through localized antagonism to soil-borne pathogens, competition for nutrients and space, and through their potential to induce plant systemic resistance (ISR) to protect plants (6, 7). *Bacillus* stands out as one of the most extensively investigated biocontrol agents, with its products reigning supreme in the biocontrol market (8). *Bacillus* is capable of forming metabolites and producing resistant endospores and also promotes plant growth through nitrogen fixation, phosphate solubilization, and phytohormone production (8, 9). The first commercial bacterial fertilizer, Alinit, was developed from *Bacillus* and increased crop yields by 40% (10). Among them, *Bacillus velezensis* can produce various secondary metabolites to benefit the plant (11). Wang et al. reported that the application of *B. velezensis* B19 in *Panax notoginseng* continuous cropping soils significantly reduced root rot in seedlings (12). Chen et al. reported that *Bacillus velezensis* WB could decrease both the incidence and the severity of watermelon wilt by modulating the structure of the microbial community in the watermelon rhizosphere (13).

In addition, plant growth-promoting rhizobacteria can play a protective role in the growth of plants under various abiotic stresses. According to relevant reports, the bacteria isolated from the roots of the halophyte *Arthrocnemum indicum* have an enhanced growth-promoting effect on peanuts under saline-alkali conditions (14). In addition, some of the properties that enable the production of cell wall-degrading enzymes are equally common mechanisms by which plant inter-root probiotics achieve disease protection.

To identify the underlying genetic traits of bacteria that promote plant growth and to gain insight into the molecular mechanisms behind biological processes, genome sequencing and its analysis have proven to be a promising approach. According to some previous reports, genes related to secondary metabolites were mined from the Q-426 genome by whole-genome sequencing and analysis of *B. velezensis* Q-426, and the production potential of the strain's secondary metabolites was analyzed in conjunction with untargeted metabolomics (15). Analysis of genes encoding antimicrobial compounds based on the whole genome sequence of *B. velezensis* FZB42 highlights the fact that microbial compounds have a biocontrol role against plant pathogens (11, 16, 17). Thus, genome-wide studies can be used to classify genes associated with the positive effects of plant growth-promoting bacteria, and complete genome sequence and metabolite analyses can help elucidate the molecular and functional mechanisms of plant growth promotion (18, 19).

In this research, our research group isolated the culturable strain D83 from the rhizosphere soil of healthy plants in a continuous cropping site of *P. notoginseng*. We have the following objectives and significance through the present study: (i) to evaluate the inhibitory activity of D83 against eight strains of phytopathogenic fungi, including *Sclerotium rolfsii*, *Fusarium oxysporum*, and *Phytophthora parasitica*, by the production of volatile gases and by the dual culture method; (ii) to evaluate the probiotic potential of D83 on plates; (iii) to evaluate the ability of D83 to promote plant growth and seed germination and compare it with common commercially available microbial agents; and (iv) to analyze the mechanism of D83 for plant growth and biocontrol through genome mining and metabolome analysis.

This study provides a research basis for the application of D83 as a PGPR and biocontrol agent in agricultural production to provide applicable solutions for sustainable agriculture.

## MATERIALS AND METHODS

### Bacterial strain

The bacterium *B. velezensis* D83 used in this study was stored in 50% (vol/vol) glycerol at −20°C and passaged on Luria-Bertani (LB) at 30°C Strain D83 was isolated and purified from the inter-root soil of healthy *P. notoginseng* in continuous cropping land in the early stage of our laboratory.

### D83 antagonistic activity against pathogenic fungi

The pathogen *S. rolfsii* has a wide host range comprising more than 500 plant species and causes Southern blight disease, which results in significant crop yield losses worldwide (20). *F. oxysporum* is the most common type of *Fusarium* (21), widespread in the environment, and capable of causing serious problems for crop production and animal or human health (22). *P. parasitica* has a hemibiotrophic lifestyle. Initially, it can invade host tissues as a biotrophic pathogen, and then, it transforms into a necrotrophic pathogen and kills the host, causing great damage to plants (23). *Fusarium graminearum* is one of the most devastating fungal pathogens worldwide (24). *Fusarium solani* causes plant wilt and root rot (25). *Fusarium fujikuroi* is one of the major phytopathogenic fungi causing rice bakanae disease worldwide (26).

This study evaluated the inhibitory capacity of isolates against eight pathogenic fungi on potato dextrose agar (PDA) using the dual culture method (27). The antagonistic ability of D83 against pathogenic fungi was determined by comparing the size of the radius of pathogenic fungi on plates inoculated only with the pathogen (control plates) and on plates inoculated with dual cultures of the pathogen and D83 (experimental plates). Three replications were set up for each treatment. Inhibition rate (100%) = (radius of control colony − radius of experimental colony)/radius of control colony × 100%. The plates were incubated at 28°C to observe the inhibitory effect on the growth of pathogenic bacteria. A plate containing only pathogenic bacteria was used as a control.

To investigate the inhibitory effect of volatile compounds produced by D83 on eight pathogenic fungi, Luria-Bertani (LB) medium and PDA were added to both sides of the separated plates, and then inoculated with D83 and pathogenic fungi on both sides, respectively, using the plate inoculated with pathogenic fungi only as a control (28). Three replications were set up for each treatment. The inhibitory capacity of volatile organic compounds (VOCs) of D83 against pathogenic fungi was calculated by measuring the radius of the experimental group and the radius of the control group. Inhibition rate (100%) = (radius of control colony − radius of experimental colony)/radius of control colony × 100%. The plates were incubated at 28°C to observe the inhibitory effect on the growth of pathogenic bacteria.

### Plant growth-promoting (PGP) attributes

To evaluate the PGP traits, the inoculum of D83 was prepared in LB overnight at 30°C (150 rpm). Siderophores are low-molecular-weight deferoxamines synthesized by microorganisms under iron-deficient conditions (29). Upon the depletion of iron in the atmospheric environment, PGPR can produce siderophores that remove iron from the surrounding environment and increase its availability to relevant hosts (30). To test the activity of siderophore production, D83 was cultured on Chrome azurol S (CAS) medium. After 3 days of incubation, the aperture around the bacterial growth on the blue medium was observed, and the presence of an orange aperture indicated the production of siderophore (31). Detection of the presence of indole-3-acetic acid (IAA)-related compounds using the Salkowski reagent (32, 33). Phosphorus solubilization capacity was

determined by inoculating D83 on Pikovaskaya agar media (HiMedia) and incubating at 30°C for 3 days. Passage of the phosphorus solubilizing capacity was determined, and the appearance of a clear halo around the colony was considered a positive reaction (34). The nitrogen-fixing ability of D83 was determined by inoculating D83 on nitrogen-free Ashby mannitol agar (35). The casein medium and 1-carboxymethylcellulase (CMC) medium were utilized on the plates to check for enzyme degradation (36, 37). D83 was cultured on an amylolytic medium to investigate the amylolytic capacity of D83.

Sodium chloride was utilized to modulate the salt concentrations of the beef extract peptone medium plates, achieving mass fractions of 3%, 5%, 8%, 11%, and 13%, respectively (38), with the aim of assessing the salt tolerance capacity of D83.

## D83's ability to promote seed germination and plant seedling growth

In the pot experiments, all the maize and tobacco seeds were purchased from the market. D83 was incubated in LB liquid medium at 30°C overnight, centrifuged at 5,000 rpm for 20 min, and the D83 precipitated product was resuspended by adding sterile distilled water and adjusting the concentration to $10^7$–$10^8$ CFU/mL. Seeds were surface sterilized by soaking in 5% sodium hypochlorite for 10 min and then rinsed six times with sterile distilled water. Surface-sterilized tobacco and maize seeds were soaked in D83 suspension and commercially available microbial preparations (effective strains *Bacillus subtilis* and *Bacillus licheniformis*) for 6 h, then placed in sterile Petri dishes lined with moistened filter papr, and germinated in a room temperature environment. The treatment using 6 h of soaking in sterile water was used as a control. Three replications were set up for each treatment. The number of germinating seeds was observed and recorded, and the filter paper was kept moist throughout the germination experiment stage. In pot experiments, 10 mL of D83 resuspension was added to each pot when the seedlings reached the four-leaf stage. Sterile distilled water and commercially available microbial preparations were used as controls. Three replicate trials were set up. Then, half of the concentration of Hoagland's nutrient was watered once a fortnight.

A total of three watering experiments with bacterial solution were set up, and the effect of D83 on seedlings was recorded by measuring plant height, root length, leaf length, biomass, fresh weight of aboveground part, and fresh weight of underground part after leaving the seedlings to grow for 2 weeks after watering. In addition, catalase (CAT), polyphenol oxidase (PPO), and malondialdehyde (MDA) were measured in the leaves of treated plants. CAT, MDA, and PPO activities were determined by using plant CAT, MDA, and PPO activity assay kits. The kits were purchased from Suzhou Grace Biotechnology Co., Ltd. The analysis was performed using a multi-mode microplate analyzer (Spectra Max iD3).

## Extraction of bacterial genomic DNA

The D83 strain was incubated in LB at 28°C for 20 h. Cells were harvested by centrifugation at 4°C, 9,391 × *g* for 30 min. The Genestar Genomic DNA Extraction Kit was used to extract genomic DNA from D83. The concentration of genomic DNA was precisely quantified by employing the Qubit 3.0 fluorometer (Life Technologies, Carlsbad, CA, USA). Subsequently, a comparative analysis was conducted between the Qubit-determined concentration and that obtained via the Nanodrop spectrophotometer, aiming to assess the purity of the samples. Finally, the integrity of genomic DNA was tested using 1.0% agarose gel electrophoresis.

## Genomic sequencing and annotation

For library creation, the DNA used should be of acceptable purity, concentration, and integrity. The fragment size of the library was detected using the Agilent 2100 Bioanalyzer (Agilent Technologies, USA). The libraries were sequenced using a PacBio Sequel II sequencer. The data were then processed using SMRT LINK 10.1.0 software. PacBio reads were reassembled using Microbial Assembly (smrtlink10), HGAP4 (39), and Canu (v.1.6)

**TABLE 1** Lipopeptide synthesis-related gene amplification primer sequences (51)

| Types of antibiotics | Primer name | Primer sequence (5′–3′) | Target gene | Amplicon size (bp) |
|---|---|---|---|---|
| Fengycin | FNDF1 | CCTGCAGAAGGAGAAGTGAAG | *fen*D | 293 |
| | FNDR1 | TGCTCATCGTCTTCCGTTTC | | |
| Fengycin | FenB1F | TACCAATCGCAATGTCGTGT | *fen*B | 767 |
| | FenB1R | CTTCGATTTCTAACAGCCGC | | |
| Unknown protein | 147F | CAGAGCGACAGCAATCACAT | *ynd*J | 212 |
| | 147R | TGAATTTCGGTCCGCTTATC | | |
| Surfactin | 110F | GTTCTCGCAGTCCAGCAGAAG | *srf*AB | 308 |
| | 110R | GCCGAGCGTATCCGTACCGAG | | |
| Iturin | ituD2F | GATGCGATCTCCTTGGATGT | *itu*D | 647 |
| | ituD2R | ATCGTCATGTGCTGCTTGAG | | |
| Iturin | bamB1F | AATTTTTCAAGCA | *itu*B | 508 |
| | bamB1R | CGACATACAGTTCTCCCGGT | | |
| Iturin | ituA1F | TGCCAGACAGTATGAGGCAG | *itu*A | 885 |
| | ituA1R | CATGCCGTATCCACTGTGAC | | |
| Fengycin | FenB1F | TACCAATCGCAATGTCGTGT | *fen*B | 767 |
| | FenB1R | CTTCGATTTCTAACAGCCGC | | |
| Iturin | ITUCF1 | TTCACTTTTGATCTGGCGAT | *itu*C | 575 |
| | ITUCR3 | CGTCCGGTACATTTTCAC | | |

(40) software. The alignment and analysis of the generated HiFi read to the assembled genome were performed using the minimap2 (v2.15-r905) tool to evaluate genome coverage depth.

The assembled genome sequence of strain D83 has been uploaded and preserved in the National Center for Biotechnology Information (NCBI) GenBank under accession number CP156684. For comparative analysis, the genomic sequences of related strains including *B. velezensis* ZF145 (CP061176), *B. velezensis* sx01604 (CP018007), *Bacillus amyloliquefaciens* GKTO4 (CP072120), and *Bacillus sp.* LUNF1 (CP048876) was downloaded from the NCBI website. These sequences were used as references to compare and analyze the genomic features and characteristics of D83.

Various software tools were employed to perform comparative analyses and gene annotation. The collinearity analysis of the strains was performed based on Mauve (v20150226) (41) software. Protein-coding sequences (CDS) were obtained using the Cluster of Orthologous Groups (COG) (42). The tRNA prediction of the D83 genome was carried out using tRNAscan-SE (2.0.9) (43). rRNA prediction was conducted using RNAmmer (v1.2) (44). For the prediction of potential rRNAs (excluding tRNA and rRNA), the genomes were matched to the Rfam database via the cmscan program of Infernal (v1.1.4) (45). The prediction of tandem repeat sequences in the D83 genome was accomplished using trf 4.09 software (46). The CRISPR structure of genomes was predicted using MinCED (v0.4.2).

## Phylogenetic analysis of D83

Prokka was employed to extract annotation information from the genomic sequence, with the resultant generation of a GFF3 file (47). Multiple sequence alignment of the D83 genome was performed using Roary (48). FastTree 2.1 (49) was employed to construct a phylogenetic tree. The construction involved performing 1,000 bootstrap replicates using the Approximate Maximum Likelihood (AML) approach. The NCBI GenBank server was used to download the genome of all strains. iTOL was utilized to enhance the visual appeal of the phylogenetic tree.

## Prediction and detection of antifungal metabolites

To identify BGCs, the genome sequence of D83 was analyzed with the antiSMASH 7.0 website, and the default settings were applied for prediction (50). To identify the genes of D83 that produce antifungal metabolites, specific amplification primers for lipopeptide antibiotic synthesis genes were designed with reference to the method of Cao et al. (51), and the primers were synthesized by Beijing Qingke Biotechnology Co. PCR amplification primers are shown in (Table 1).The PCR amplification of lipopeptide antibiotic synthesis genes was carried out using D83 genomic DNA with good concentration, purity, and integrity. The 25 μL reaction system was: 2 × SuperNova PCR Mix (Dye) 12.5 μL, 2 μL R and F, genomic DNA 1 μL, ddH$_2$O 7.5 μL.

The PCR reaction conditions for amplification of *fen*D, *itu*C, *ynd*J, and *itun*A were as follows: pre-denaturation at 98°C for 3 min, denaturation at 98°C for 10 s, annealing at 52°C for 30 s, extension at 72°C for 30 s, 35 cycles, and additional extension at 72°C for 5 min. The PCR reaction conditions for amplification of *sur*AB and *itu*D were as follows: pre-denaturation at 98°C for 3 min, denaturation at 98°C for 10 s, annealing at 58°C for 30 s, extension at 72°C for 30 s, 35 cycles, and additional extension at 72°C for 5 min. The PCR reaction conditions for amplification of *fen*B and *itu*B were as follows: denaturation at 98°C for 10 s, annealing at 56°C for 30 s, extension at 72°C for 30 s, 35 cycles, and supplementary extension at 72°C for 5 min. The PCR product bands were detected using a 1.0% agarose gel.

## LC/MS analysis of D83

We extracted the metabolites produced by *B. velezensis* D83, named D83. The samples were dispatched to Wuhan Fraser Genetic Information Co., Ltd. for untargeted metabolomics analysis. The experimental conditions were as follows.

The samples were analyzed on a Waters ACQUITY Premier HSS T3 Column (2.1 mm × 100 mm, 1.8 μm) coupled to an LC - 30A ultra-high-performance liquid chromatograph (Japan). The analysis was conducted at a flow rate of 0.4 mL/min and a column temperature of 40°C. During the entirety of the analytical process, the samples were consistently maintained at a temperature of 4°C within the autosampler, thereby ensuring the stability and integrity of the samples for subsequent analysis. The chromatographic conditions included the utilization of two mobile phases: (A) 0.1% formic acid in water

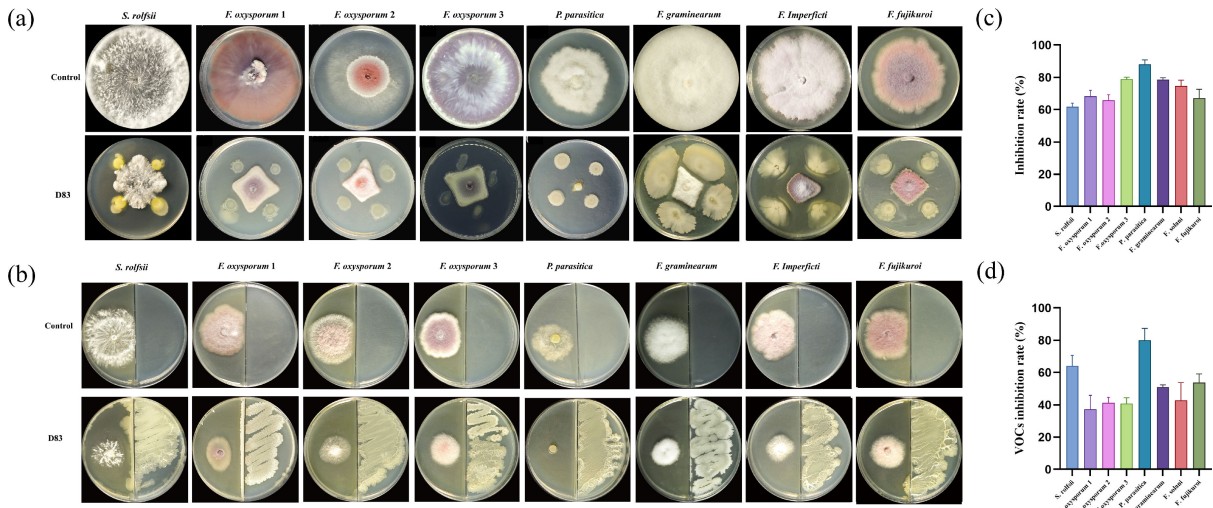

**FIG 1** Antagonistic ability of strain D83 against plant fungal pathogens. (**a**) Control: Plates inoculated only with phytopathogenic fungi. Plates inoculated with plant fungal pathogens alone; D83: Plates inoculated with both D83 and fungal pathogens. (**b**) Control: Two-compartment plates inoculated with fungal pathogens on one side only; D83: One side of the two-grid plate was inoculated with D83 and the other side with the fungal pathogen. (**c**). The inhibitory capacity of strain D83 against eight fungal pathogens in a dual culture assay (**d**). D83 Inhibitory capacity of volatile gases produced against eight fungal pathogens. From left to right, *S. rolfsii*, *F. oxysporum* 1, *F. oxysporum* 2, *F. oxysporum* 3, *P. parasitica*, *F. graminearum*, *F. solani*, and *F. fujikuroi*.

and (B) 0.1% formic acid in acetonitrile. The flow rate during chromatography was set at 0.4 mL/min. The procedures were: 0–2 min, gradient 95%–80% A, 2–5 min, gradient 80%–40% A, 5–6 min, gradient 40%–1% A, 6–7.5 min, 99% B, 7.5–7.6 min, gradient 99%–5% B, and 7.6–10 min, 5% B. The chromatographic conditions were as follows.

Mass spectrometry analysis was performed using a TripleTOF 6600 + mass spectrometer (Foster City, CA, USA). The mass spectrometry conditions were as follows: electrospray ionization (ESI), ion source temperature: 550°C, ion spray voltage: 5,000 V, spray gas: 50 psi, curtain gas: 35 psi, auxiliary heated hot gas: 60 psi, de-clustering voltage: 60 V, MS1 collision energy: 10 V, and MS2 collision energy: 30 V. The mass spectrometry was performed on a TripleTOF 6600 + mass spectrometer (Foster City, CA, USA).

## Data analysis

Statistical analysis was performed by one-way analysis of variance (ANOVA) using GraphPad Prism 10 software. Data were shown as mean ± standard deviation (SD) of three independent biological replicates.

## RESULTS AND DISCUSSION

### Antagonistic ability of strain D83 against pathogenic fungi

Plate antagonism experiments showed that strain D83 inhibited all eight fungal pathogens, as evidenced by the inhibition of pathogen growth by D83. Inhibition rates ranged from 61.79% to 88.11% (Fig. 1a and c). The VOCs plate also showed the inhibitory effect of D83 on eight fungal pathogens, which was also demonstrated by the inhibition of the growth of pathogenic bacteria by the volatile gas produced by D83. The inhibition

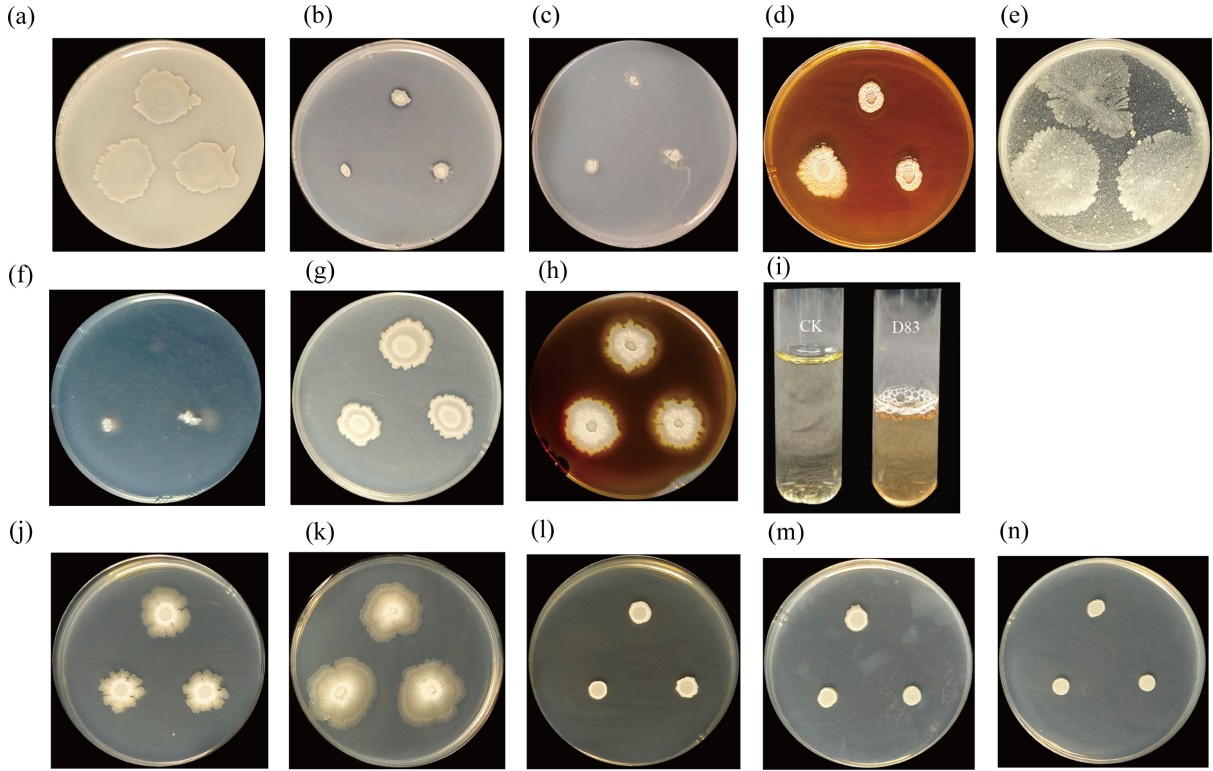

FIG 2 Growth-promoting properties of strain D83. (**a**) Ability to dissolve inorganic phosphorus. (**b**) The ability to dissolve organic phosphorus. (**c**) Nitrogen fixation capacity. (**d**) Cellulase production capacity. (**e**) Ability to dissolve potassium. (**f**) Siderophore production capacity. (**g**) The ability of proteases to produce. (**h**) Amylase production capacity. (**i**) Capacity for indoleacetic acid production. (**j**) Three percent sodium chloride concentration. (**k**) Five percent sodium chloride concentration. (**l**) Eight percent sodium chloride concentration. (**m**) Eleven percent sodium chloride concentration. (**n**) Thirteen percent sodium chloride concentration.

rate ranged from 37.25% to 79.88% (Fig. 1b and d). This suggests that strain D83 has broad-spectrum antimicrobial resistance. It is also worth noting that plate antagonism and VOCs showed different trends in inhibitory capacity, but both inhibited the growth of eight fungal pathogens. This suggests that D83 may depend on a different mechanism of action when it comes to antimicrobial activity. In summary, *B. velezensis* D83 has broad-spectrum resistance to bacteria and has the potential to control a wide range of plant soil-borne diseases. In addition, volatile gases produced by D83 are potential biocontrol mechanisms for pathogenic fungi.

In this study, we isolated *B. velezensis* D83 from the inter-root soil of healthy *P. notoginseng* grown continuously. The D83 exhibited broad-spectrum antimicrobial activity against *S. rolfsii, F. oxysporum, P. parasitica, F. graminearum, F. solani*, and *F. fujikuroi in vitro*. This suggests that D83 has the potential to suppress plant diseases in agricultural production.

## Plant growth-promoting properties of strain D83

Plate experiments demonstrated that D83 possesses the capability to dissolve inorganic and organic phosphorus; fix nitrogen and cellulase production capacity; and produce siderophore, protease, amylase, and indoleacetic acid (Fig. 2a through i). Furthermore, the growth of strain D83 was observed on sodium chloride plates with varying concentrations, and it still survived at a 13% sodium chloride concentration, suggesting that D83 has good salt tolerance (Fig. 2j through n). These growth attributes suggest that strain D83 is a potential plant growth-promoting bacterium. It also has the potential for application in salinized soils.

## D83 promotes plant seedling growth and seed germination

*B. velezensis* D83-treated corn seeds and tobacco seeds had the highest germination rates of 95% and 85%, respectively. In addition, seeds treated with D83 also germinated faster than the sterile water treatment and the A treatment group (Fig. 3).

The growth-promoting activity of D83 was further confirmed by the pot test. Compared with the control, D83 had a good effect on the growth of maize seedlings. Specifically, D83 significantly ($P < 0.05$) increased the leaf length, root length, biomass, fresh weight of underground parts, and fresh weight of aboveground parts of maize seedlings. Although commercially available microbial preparations also increased the

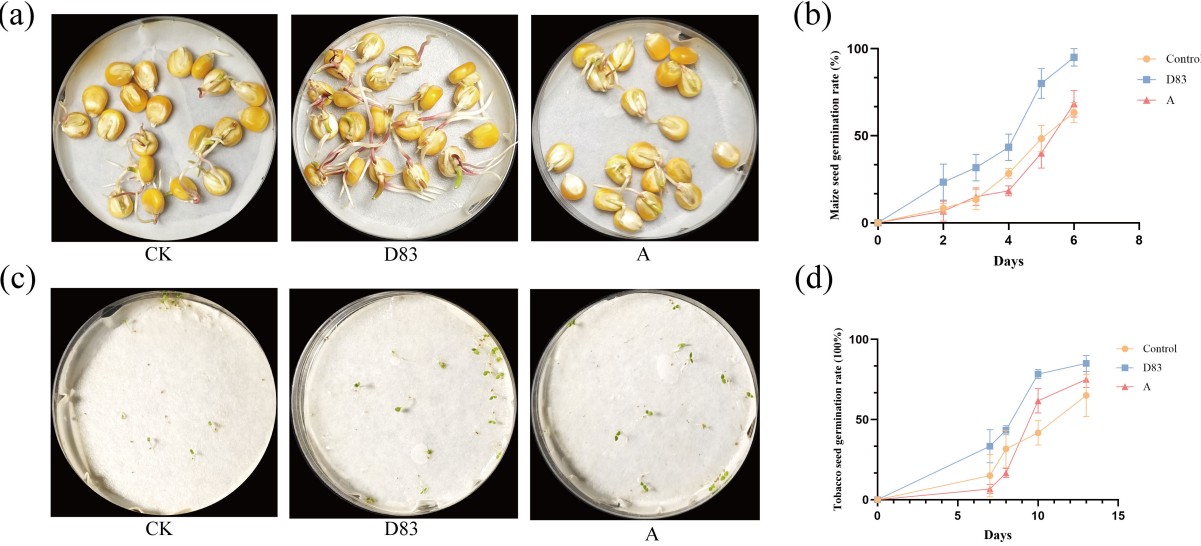

**FIG 3** Effect of D83 on seed germination. (**a**) *In vitro* germination experiments with maize seeds. (**b**) Maize seed germination rate. (**c**) *In vitro* germination of tobacco seeds. (**d**) Tobacco seed germination rate. D83: Seeds treated with *B. velezensis* D83 suspension; CK: Seeds treated with sterile water; A: Seeds treated with commercially available microbial preparations.

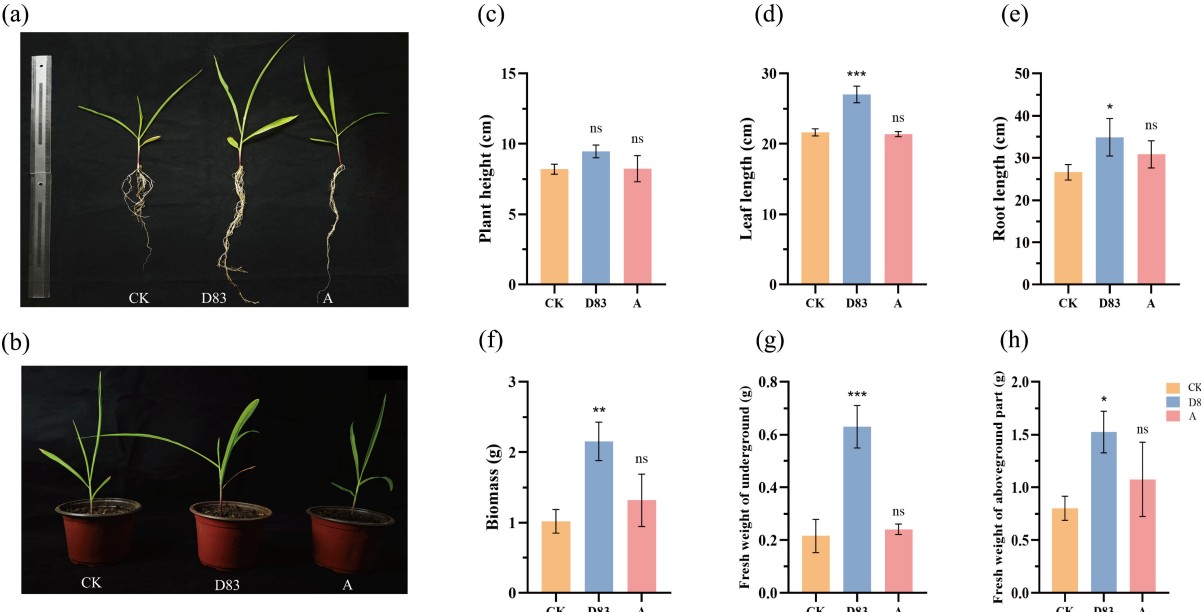

**FIG 4** Influence of *B. velezensis* D83 as soil inoculant on the growth of maize in pot experiments. (**a**) and (**b**) Treated maize plants. D83: Maize seedlings treated with *B. velezensis* D83 suspension; CK: Maize seedlings treated with sterile water; A: Maize seedlings treated with commercially available microbial preparations. (**c**) Plant height (ns $P$ = 0.0849; ns $P$ = 0.9969). (**d**) Leaf length (***$P$ = 0.0003; ns $P$ = 0.9082). (**e**) Root length (*$P$ = 0.0388; ns $P$ = 0.2758). (**f**) Biomass (**$P$ = 0.0048; ns $P$ = 0.3853). (**g**) Fresh weight of underground part (***$P$ = 0.0003; ns $P$ = 0.8352). (**h**) Fresh weight of aboveground part (*$P$ = 0.0188; ns $P$ = 0.3421). Data represent mean ± SD of three biological replicates. Asterisks indicate significant differences.

mean values of maize seedling traits. However, this effect was not significant (Fig. 4). Similarly, plant height, leaf length, root length, biomass, fresh weight of underground parts, and fresh weight of aboveground parts were significantly ($P$ < 0.05) increased in tobacco seedlings inoculated with D83 suspension in soil compared with control. In addition, D83 was more effective in promoting tobacco growth than commercially available microbial formulations (Fig. 5).

In pots where D83 was applied as a soil inoculant, the growth of tobacco seedlings and maize seedlings was significantly better than that of the sterile water control and commercially available microbial preparations. The growth of the plant was significantly better than that of the sterile water control and commercially available microbial preparations. The growth of the plant was significantly better than that of the sterile water control and commercially available microbial preparations. This suggests that strain D83 has the potential to be used as a plant growth promoter, promotes the growth of a wide range of plants, and increases plant production.

## Effect of D83 on plant enzyme activities

D83 treatment increased catalase and polyphenol oxidase activity and decreased MDA content in maize plants compared with control, but none of them were significant. In contrast, the A treatment significantly reduced the hydrogen oxidase activity. A treatment also increased MDA content and PPO activity, but both were not significant. In addition, D83 treatment significantly increased CAT and PPO activity in tobacco plants and decreased MDA content (Fig. 6).

PPO plays an important role in plant immune response, abiotic stress, and physiological metabolism (52). MDA is a widely used marker of oxidized lipid damage, reflecting the degree of cell membrane lipid peroxidation and the strength of the plant response to adverse conditions. CAT is a common and highly active enzyme in living organisms, and its overexpression increases plant resistance to abiotic and biotic stresses. The increase in PPO and CAT activities and decrease in MDA content after D83 treatment may indicate that D83 enhances the immune response of plants under stress.

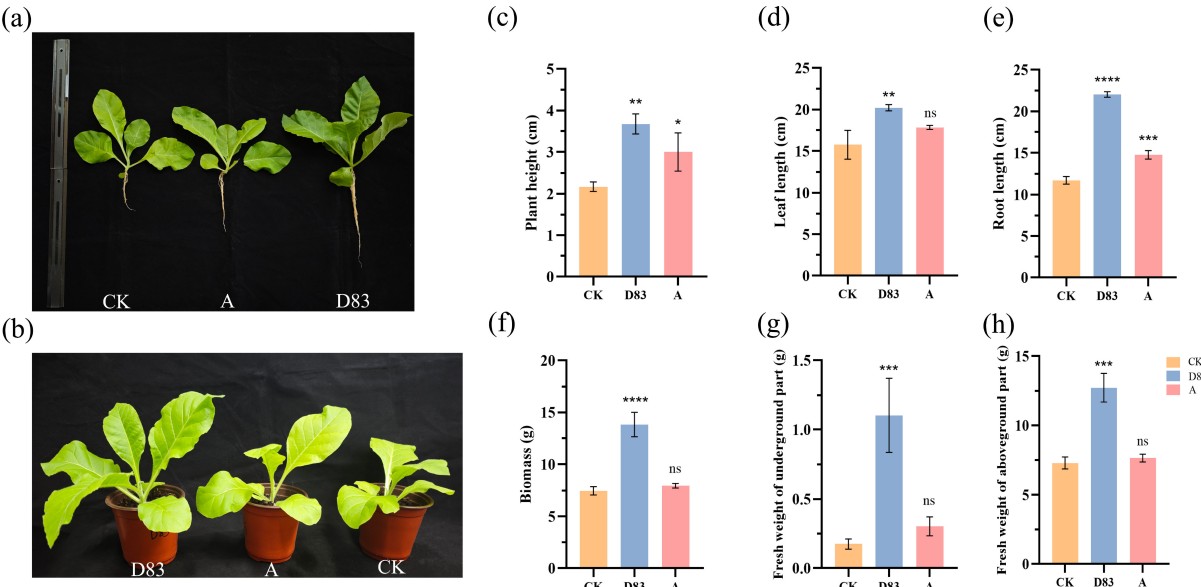

**FIG 5** Influence of *B. velezensis* D83 as soil inoculant on growth of tobacco in pot experiments. (**a**) and (**b**) treated tobacco plants. D83: Tobacco seedlings treated with *B. velezensis* D83 suspension; CK: tobacco seedlings treated with sterile water; and A: tobacco seedlings treated with commercially available microbial preparations. (**c**) Plant height (**$P = 0.0017$; *$P = 0.0278$). (**d**) Leaf length (**$P = 0.0033$; ns $P = 0.0278$). (**e**) Root length (****$P < 0.0001$; ***$P = 0.0002$). (**f**) Biomass (****$P < 0.0001$; ns $P = 0.6484$). (**g**) Fresh weight of underground part (***$P = 0.0007$; ns $P = 0.5547$). (**h**) Fresh weight of aboveground part (***$P = 0.0001$; ns $P = 0.07475$). Data represent mean ± SD of three biological replicates. Asterisks indicate significant differences.

The germination rate of maize and tobacco seeds was significantly increased after treatment with D83. In addition, seedlings of maize and tobacco treated with D83 showed faster growth, higher CAT and PPO activity, and lower MDA content than seedlings treated with sterile water and commercially available microbial preparations, demonstrating that D83 has potential as a plant growth regulator.

## Phylogenetic analysis based on the whole genome of D83

A phylogenetic analysis based on the whole-genome sequence was performed on strain D83, and its phylogenetic relationship with related *B. velezensis* was analyzed. Based on the phylogeny, D83 formed a separate monophyletic group with *B. velezensis* FZB42, *B. velezensis* AP46, *B. velezensis* DSYZ, *B. velezensis* GJJK74, and *B. velezensis* 11640. The above branch shares a node with *B. velezensis* 160. Based on the phylogenetic relationship, D83 was identified as *B. velezensis* D83 (Fig. 7a). In this branch, most of the *B. velezensis* strains have been reported for their plant growth-promoting and biocontrol traits, such as *B. velezensis* FZB42 (16) and *B. velezensis* DSYZ (53).

## D83 Basic features of the genome

Strain D83 had a 3,929,757 bp circular chromosome with an average G + C content of 46.50% (GenBank accession number CP156684). A total of 3,987 protein-coding genes, 27 rRNA (16 S-23S-5S rRNA) genes, and 86 tRNA genes were predicted in the genome (Fig. 7b and Table 2). In addition, there were eight clustered regularly interspaced short palindromic repeat sequences (CRISPRs).

Among the 4,020 protein-coding genes, 3,987 (99.18%) were annotated with predicted function. The genomic sequences were compared with five commonly used databases, and 3,987, 3,510, 3,111, 2,245, and 3,077 genes were matched to sequences in the non-redundant protein database (NR), Swiss-Prot, COG, Kyoto Encyclopedia of Genes and Genomes (KEGG), and gene ontology (GO) databases, respectively (Fig. 8a). Furthermore, 3,111 genes were classified into 24 COG categories, and most of them were associated with functions such as amino acid transport and metabolism, carbohydrate

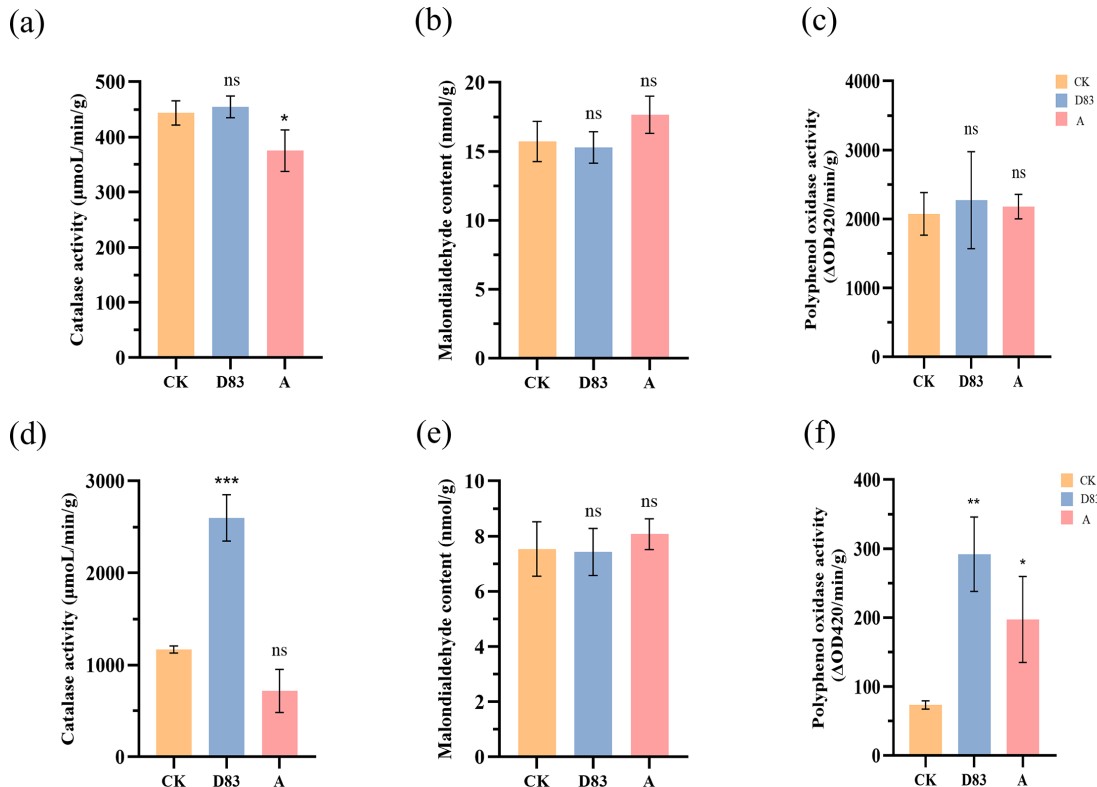

**FIG 6** (**a**) Effect of D83 treatment on CAT activity in maize plants (ns $P = 0.0849$; *$P = 0.0040$). (**b**) Effect of D83 treatment on MDA content in maize plants (ns $P > 0.05$). (**c**) Effect of D83 treatment on PPO activity in maize plants (ns $P > 0.05$). (**d**) Effect of D83 treatment on CAT activity in tobacco plants (***$P = 0.0002$; ns $P = 0.0573$). (**e**) Effect of D83 treatment on MDA content in tobacco plants (ns $P > 0.05$). (**f**) Effect of D83 treatment on PPO activity in tobacco plants (**$P = 0.0025$; *$P = 0.0335$). D83: Seedlings treated with *B. velezensis* D83 suspension; CK: Seedlings treated with sterile water; and A: Seedlings treated with commercially available microbial preparations.

transport and metabolism, transcription, signal transduction mechanisms, and secondary metabolites biosynthesis (Fig. 8b).

## Comparative analysis of the genomes of D83 and four other strains

The GC content of *B. velezensis* D83 was not significantly different from that of the other four strains, and the genome was smaller than that of *B. amyloliquefaciens* GKT04 and *Bacillus* sp LUNF1, and not much different in size from the other two strains. The number of genes encoding proteins in D83 was 3,987, which was higher than the number of genes encoding proteins in *B. velezensis* ZF145 (3,628), *B. velezensis* sx01604 (3,672), *Bacillus sp* LUNF1 (3,951), and *B. amyloliquefaciens* GKT04 (3,936), which may indicate that *B. velezensis* D83 has the potential to encode more proteins than other *Bacillus* strains. (Table 2).

To assess the genomic similarity and variation among different strains, in this study, genome-wide collinearity analysis was performed on several *Bacillus* strains with an outgroup (*Staphylococcus aureus* NCTC 8325) using MAUVE software. D83 is located at the top, and its LCB alignment is largely complete and continuous. In comparison with the other strains, D83 shows regions of collinearity that are broadly consistent with those in the other strains, suggesting that they are evolutionarily closely related. Most LCBs were arranged in a similar order across multiple strains, suggesting that these strains have a highly conserved genomic structure. Despite the LCBs of these strains being highly similar to D83, some genomic rearrangement events, such as inversions, insertions, and deletions, were observed in certain regions (Fig. 7c).

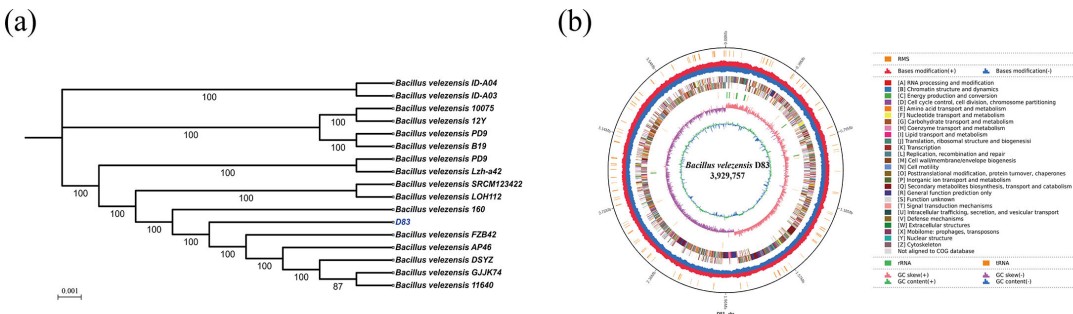

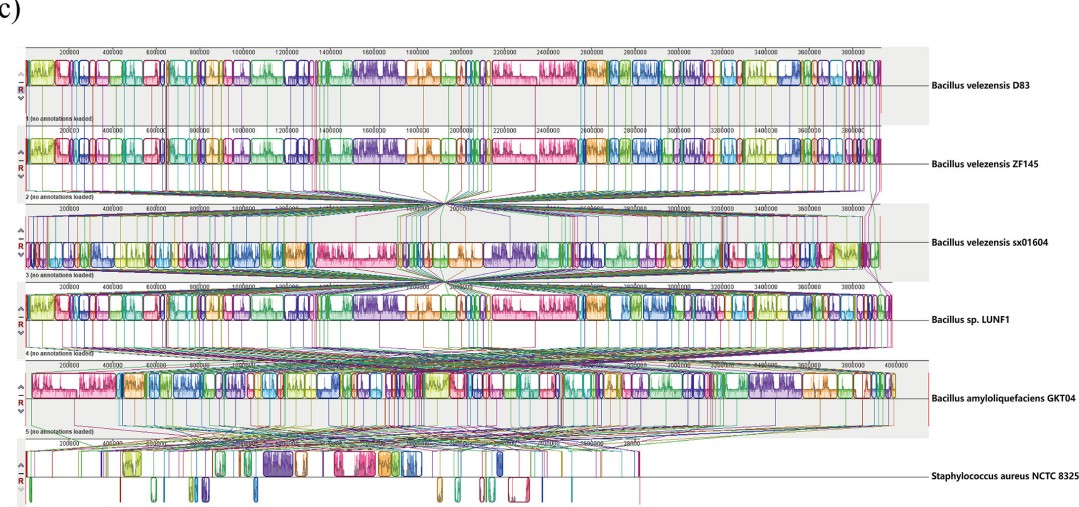

**FIG 7** (**a**) The phylogenetic tree constructed based on the whole genome of D83. The value at each node corresponds to bootstrap values. (**b**) Circular genome map of *B. velezensis* D83. From the innermost to the outermost: ring 1 for GC content, ring 2 for GC skew, ring 3 for distribution of rRNAs (green) and tRNAs (brown), ring 4 for COG classifications of protein-coding genes on the forward strand and reverse strand, ring 5 for restriction-modification system, and ring 6 for genome size (black line). (**c**) Comparative genomic analysis of the genomes of D83 and five reference strains. The colored blocks shown in the figure are local collinear blocks (LCBs), which represent sequentially arranged regions conserved across multiple genomes.

## Genetic basis for plant growth promotion

Genome analysis revealed the presence of gene clusters for the biosynthesis of plant promotion and antifungal compounds, such as IAA, tryptophan, 3-hydroxy-2-butanone, 2,3-butanediol, siderophores, spermidine, and phytase. (Table 3). Two common volatile organic compounds, 3-hydroxy-2-butanone and 2, 3-butenediol, not only induce systemic resistance to pathogens in plants but also promote plant growth (56). Genes responsible for the synthesis of 3-hydroxy-2-butanone, including acetolactate synthase (*als*S), acetolactate decarboxylase (*als*D), acetolactate synthase large subunit (*ilv*B), and acetolactate synthase small subunit (*ilv*H), are found in the D83 genome (Table 3).

**TABLE 2** Comparison of *B. velezensis* D83 with four other strains of *Bacillus* sp

| Features | *B. velezensis* D83 | *B. velezensis* ZF145 | *B. velezensis* sx01604 | *Bacillus sp* LUNF1 | *B. amyloliquefaciens* GKT04 |
|---|---|---|---|---|---|
| Genome size (bp) | 3,929,757 | 3,929,800 | 3,926,520 | 3,980,777 | 4,056,188 |
| GC content (%) | 46.5 | 46.5 | 46.5 | 46.46 | 46.39 |
| Protein-coding genes | 3,987 | 3,628 | 3,672 | 3,951 | 3,936 |
| Numbers of tRNAs | 86 | 86 | 86 | 86 | 86 |
| rRNA | 26 | 27 | 27 | 27 | 27 |
| Accession no. | NZ_CP156684 | NZ_CP061176 | NZ_CP018007 | NZ_CP048876 | NZ_CP072120 |
| Reference | This Study | Unpublished | Unpublished | (54) | (55) |

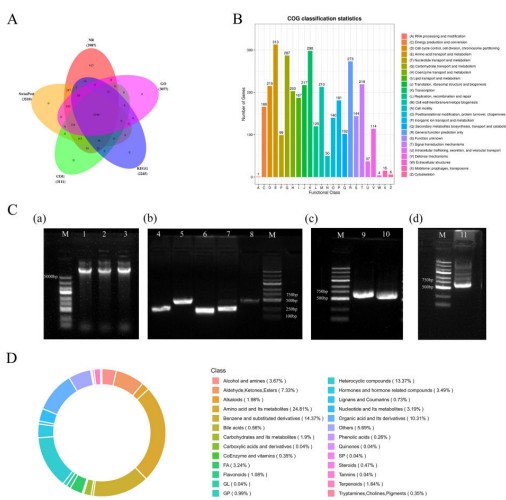

**FIG 8** (A) Venn diagram of *B. velezensis* D83 based on NR, Swiss-Prot, COG, KEGG, and GO databases. (B) COG annotation of *B. velezensis* D83. (C) Agarose gel electrophoresis of PCR products of genes related to the synthesis of D83 antimicrobial substances. M: DL 2000 DNA maker; 1-3 are D83 genomic DNA bands; 4-11 are *fen*D, *itu*C, *ynd*J, *srf*AB, *itu*D, *fen*B, *itun*A, and *itu*B amplicon bands, respectively. (D) Ring diagram of metabolite class composition of D83.

Meanwhile, the gene encoding 2, 3-butanediol dehydrogenase (*bdh*A), which catalyzes the formation of 2, 3-butanediol, was found (Table 3). Phytase is a nutrient-promoting enzyme that hydrolyzes phytic acid and enhances associated nutrients to be bioavailable (57). Genes responsible for the synthesis of phytase have been identified in the D83 genome, including 3-phytase (*phy*). Spermidine was shown to play important roles in plant growth and abiotic stress responses (58). In addition, genes responsible for the synthesis of spermidine, including arginine decarboxylase (*spe*A), agmatinase (*spe*B), and spermidine synthase (*spe*E), are found (Table 3).

## Analysis of the secondary metabolite biosynthesis gene cluster of D83

With advances in sequencing technology and the development of powerful genome mining tools, more options are available for the study of secondary metabolites. The

**TABLE 3** Genes related to plant growth promotion in the D83 genome

| Gene | Gene product | Function |
| --- | --- | --- |
| *dhb* cluster | Bacillibactin | Siderophore synthesis |
| *als*S | Alpha-acetolactate synthase | 3-hydroxy-2-butanone synthesis |
| *als*D | Alpha-acetolactate decarboxylase | |
| *ilv*B | Acetolactate synthase large subunit | |
| *ilv*H | Acetolactate synthase, small subunit | |
| *bdh*A | Acetoin reductase /2,3-butanediol dehydrogenase | 2,3-butanediol synthesis |
| *phy*C | 3-phytase | Phytase synthesis |
| *ysn*E | N-acetyltransferase | IAA synthesis |
| *trp*A | Tryptophan synthase alpha subunit | |
| *trp*B | Tryptophan synthase beta subunit | |
| *trp*C | Indole-3-glycerol-phosphate synthase | |
| *trp*E | Anthranilate synthase | |
| *yhc*X | Putative amidohydrolase | |
| *spe*A | Arginine decarboxylase | Spermidine synthesis |
| *spe*B | Agmatinase | |
| *spe*E | Spermidine synthase | |

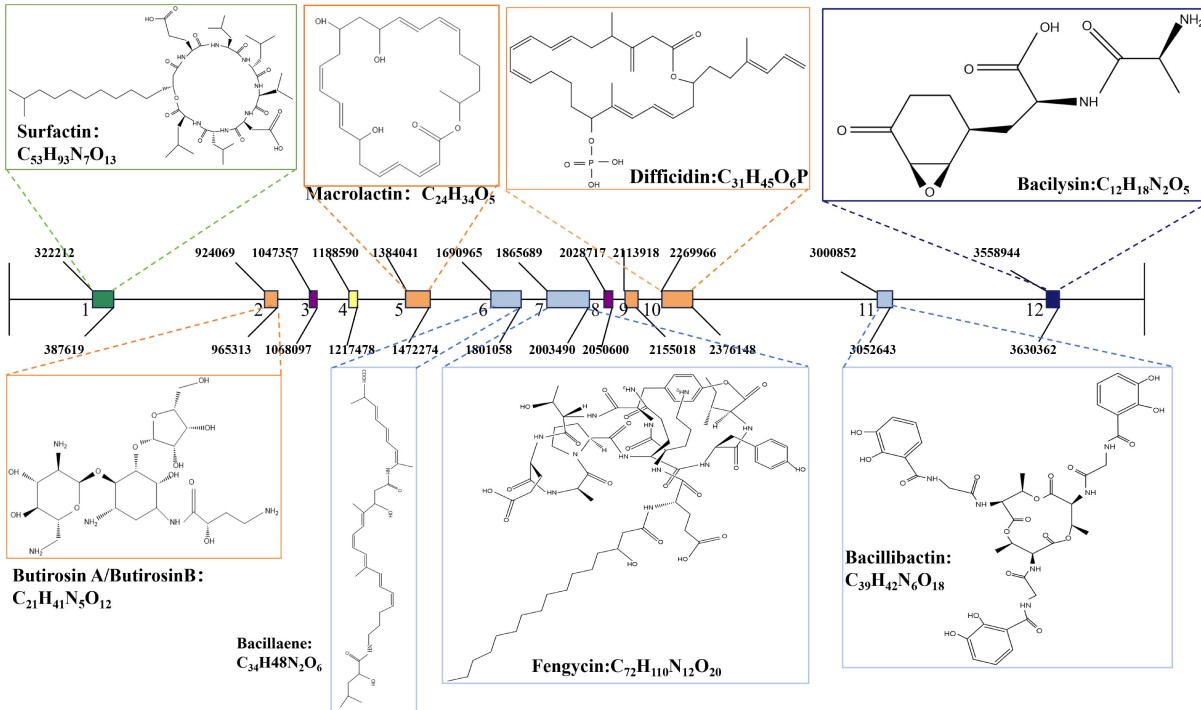

**FIG 9** Secondary metabolite gene clusters identified in *B. velezensis* D83 genome using antiSMASH software version 7.0.

antibiotics and secondary metabolite analysis shell (antiSMASH) enables rapid annotation and analysis of secondary metabolite biosynthesis gene clusters and helps estimate the types of compounds encoded by gene clusters (50, 59). *B. velezensis* has been reported to have an impressive ability to produce secondary metabolites with antimicrobial activity, said secondary metabolites include lipopeptides (surfactin, fengycin, and bacillomycin D), polyketides (macrolide, bacillaene, and difficidin/oxydifficidin), and peptides (plantazolicin, amylocyclicin, and bacilysin).

Using antiSMASH 7.0 to predict secondary metabolite biosynthesis gene clusters of *B. velezensis* D83, a total of 12 secondary metabolite clusters were analyzed and identified (Fig. 9 and Table 4). Among them, six clusters have 100% nucleotide sequences homology with known gene clusters, including transAT-PKS (cluster 5 and cluster 10), transAT-PKS, T3PKS, NRPS (cluster 6), NRPS, transAT-PKS, betalactone (cluster 7), NRP-metallophore, the NRPS, RiPP-like (cluster 11), and others (cluster 12). However, cluster 1, cluster 2, cluster 5, cluster 6, cluster 7, cluster 10, cluster 11, and cluster 12 were associated with the production of surfactin, butirosin A/B, macrolactin H, bacillaene, fengycin, difficidin, bacillibactin, and bacilysin, respectively. The structures of these secondary metabolites are shown in Fig. 9.

Furthermore, comparing D83 with four *Bacillus* strains, Table 4 shows that the clusters of secondary metabolite genes were highly similar among the five strains (Table 5). All five strains had clusters of genes encoding surfactin, macrolactin H, bacillaene, fengycin, difficidin, bacillibactin, and bacilysin. In addition, among these five strains, D83 alone had the cluster of genes encoding butirosin A/B. *B. amyloliquefaciens* GKT04 alone had the cluster of genes encoding plantazolicin.

Non-ribosomal enzyme-synthesized lipopeptide antibiotics are the most common class of antimicrobial substances such as fengycin, iturin, and surfactin (60). Figure 9 shows D83 genomic DNA with good integrity. Lanes 4–11 show PCR amplicons for *fen*D, *itu*C, *ynd*J, *srf*AB, *itu*D, *fen*B, *itun*A, and *itu*B, respectively, and the size of the PCR products matched the theoretical values indicating the presence of these genes (Fig. 8c). The above genes encode fengycin, iturin, and surfactin, respectively.

**TABLE 4** Cluster of genes encoding secondary metabolites in the genome of *B. velezensis* D83[a]

| Cluster number | Cluster name | From | To | Most similar Known cluster | | Similarity |
|---|---|---|---|---|---|---|
| 1 | NRPS | Chr1:322,212 | Chr1:387,619 | Surfactin | NRP:Lipopeptide | 82% |
| 2 | PKS-like | Chr1:924,069 | Chr1:965,313 | Butirosin A/B | Saccharide | 7% |
| 3 | Terpene | Chr1:1,047,357 | Chr1:1,068,097 | – | – | – |
| 4 | Lanthipeptide-class-ii | Chr1:1,188,590 | Chr1:1,217,478 | – | – | – |
| 5 | TransAT-PKS | Chr1:1,384,041 | Chr1:1,472,274 | Macrolactin H | Polyketide | 100% |
| 6 | TransAT-PKS, T3PKS, NRPS | Chr1:1,690,965 | Chr1:1,801,058 | Bacillaene | Polyketide + NRP | 100% |
| 7 | NRPS, transAT-PKS, betalactone | Chr1:1,865,689 | Chr1:2,003,490 | Fengycin | NRP | 100% |
| 8 | Terpene | Chr1:2,028,717 | Chr1:2,050,600 | – | – | – |
| 9 | T3PKS | Chr1:2,113,918 | Chr1:2,155,018 | – | – | – |
| 10 | TransAT-PKS | Chr1:2,269,966 | Chr1:2,376,148 | Difficidin | Polyketide | 100% |
| 11 | NRP-metallophore, NRPS, RiPP-like | Chr1:3,000,852 | Chr1:3,052,643 | Bacillibactin | NRP | 100% |
| 12 | Other | Chr1:3,558,944 | Chr1:3,630,362 | Bacilysin | Other | 100% |

[a]"–" indicates information that has not been annotated.

## Analysis of secondary metabolites in *B. velezensis* D83

To gain a colonization advantage over competing bacteria, PGPR secretes a series of secondary metabolites to inhibit receptor bacteria, thereby indirectly stimulating plant growth. We identified a substantial number of antimicrobial metabolites (2,648) in strain D83 through untargeted metabolomics (Table S1). A significant portion of these metabolites possesses potential functions. For instance, Taxol C, Natamycin, Enoxacin, and Kanamycin are among them, and many of these compounds have been used in various antibiotics, antifungal pesticides, and antitumor agents (61–64).

In addition, we detected indoleacetic acid production in metabolites. Indoleacetic acid production was also observed in growth promotion experiments with strain D83 (Fig. 8D). This indicates that the strain has the capability of producing indoleacetic acid. However, we neither detected the metabolites, nor detected the metabolites surfactin, butirosin A/butirosin B, macrolactin H, bacillaene, fengycin, difficidin, bacillibactin, and bacilysin, encoded by the cluster of secondary metabolite synthesizing genes based on the whole genome prediction of D83. It may be due to strain culture conditions that lack the precursors needed to synthesize these substances or the silencing of the gene cluster transcription factors for synthesis. The possibility of synthesizing these metabolites can be subsequently increased by adding precursor substances or activating transcription factors.

**TABLE 5** Comparison of predicted and known secondary metabolites between *B. velezensis* D83, *B. velezensis* ZF145, *B. velezensis* sx01604, and *B. amyloliquefaciens* GKT04

| Cluster | *B. velezensis* D83 | *B. velezensis* ZF145 | *B. velezensis* sx01604 | *Bacillus sp* LUNF1 | *B. amyloliquefaciens* GKT04 |
|---|---|---|---|---|---|
| 1 | Surfactin | Surfactin | Bacilysin | Surfactin | Difficidin |
| 2 | Butirosin A/B | Unknown 1 | Bacillibactin | Unknown 1 | Unknown 1 |
| 3 | Unknown 1 | Unknown 2 | Unknown 1 | Unknown 2 | Bacillibactin |
| 4 | Unknown 2 | Unknown 3 | Difficidin | Unknown 3 | Bacilysin |
| 5 | Macrolactin H | Macrolactin H | Unknown 2 | Macrolactin H | Surfactin |
| 6 | Bacillaene | Bacillaene | Unknown 3 | Bacillaene | Plantazolicin |
| 7 | Fengycin | Fengycin | Fengycin | Fengycin | Unknown 2 |
| 8 | Unknown 3 | Unknown 4 | Bacillaene | Unknown 4 | Unknown 3 |
| 9 | Unknown 4 | Unknown 5 | Macrolactin H | Unknown 5 | Macrolactin H |
| 10 | Difficidin | Difficidin | Unknown 4 | Difficidin | Bacillaene |
| 11 | Bacillibactin | Unknown 6 | Unknown 5 | Unknown 6 | Fengycin |
| 12 | Bacilysin | Bacillibactin | Unknown 6 | Bacillibactin | Unknown 4 |
| 13 | – | Bacilysin | Surfactin | Bacilysin | Unknown 5 |

## Conclusion

Plant growth-promoting rhizobacteria (PGPR) inhibits pathogens and promotes plant growth and may also produce antibiotics, compete with pathogens for nutrients, or induce systemic resistance in host plants against pathogens (65). In this study, *B. velezensis* D83, which was isolated from the inter-root soil of healthy *P. notoginseng* in a continuous cropping field, showed strong antagonistic activity against the strains of *Phytophthora nicotianae*, *F. oxysporum*, *P. parasitica*, *F. graminearum, F. solani*, and *F. fujikuroi* (Fig. 1). In addition, strain D83 has good plant growth-promoting ability and high salt tolerance (Fig. 2). The above results indicate that *B. velezensis* D83 has a wide range of potential applications in the biological control of plant diseases, promotion of crop growth, and growth under abiotic stress conditions. We confirmed the growth-promoting effect of D83 on plants through *in vitro* potting experiments and seed germination experiments (Fig. 3 to 5). This growth-promoting effect was significant and better than that of the commercially available microbial preparations treatment, which also indicates the great potential of the D83 formulation for agricultural production applications.

Whole genome sequencing analysis of strain D83 uncovered 12 clusters and 540 genes encoding secondary metabolites with predicted functions, including the surfactin and fengycin families (Fig. 9). Analysis of PCR amplicon results based on fengycin, surfactin, and iturin-specific primers showed that D83 has the genes to synthesize fengycin, surfactin, and iturin (Fig. 9c). In addition, through D83 genomic information, we identified genes that promote plant growth.

Metabolites with antifungal activity, growth-promoting, antibiotic, and antitumor effects were detected in the fermentation broth of strain D83 using LC-MS/MS, suggesting that strain D83 is a potentially usable strain resource with the potential to produce abundant secondary metabolites (Fig. 9d). In summary, strain D83 can be developed as a plant growth promoter with potential for promoting plant growth, resisting plant diseases, and being applied in saline soils. However, to ascertain the applicability of *B. velezensis* D83 as a plant growth promoter in agricultural settings, field trials are requisite to validate its efficacy in the field. Furthermore, we predicted that *B. velezensis* D83 has substantial potential to produce a wide array of secondary metabolites. This work contributes to our understanding of its ability to biosynthesize secondary metabolites and lays the foundation for future exploitation of this valuable resource.

These results indicate that *B. velezensis* D83 has a wide range of potential applications in biological control of plant diseases, promotion of crop growth, and growth under abiotic stress conditions.

## ACKNOWLEDGMENTS

This study was financially supported by the project of The Sino-Vietnamese International Joint Laboratory for Characteristic & Cash Crops Green Development of Yunnan Province (202403AP140013), the Development and Application of Quality Standards for Decomposed Agricultural Fertilizer for Tobacco Use (KMYC202313), Yunnan Tobacco Chemical Key Laboratory of Yunnan Tobacco Industry Co., Ltd. (2024539200340047), and Yunnan Provincial Department of Education Scientific Research Fund Project (2024Y261).

T.L. and X.Z. conceptualized the experiment and project management. J.C. and J.D. wrote the original manuscript and genome analysis. Z.X. and D.J. revised part of the paper and completed some experiments. J.Y. collated data and formal analysis. L.H.T. and H.T.T. reviewed and edited. All authors reviewed the manuscript.

## AUTHOR AFFILIATIONS

[1]College of Agronomy and Biotechnology, Yunnan Agricultural University, Kunming, China

[2]National-Local Joint Engineering Research Center on Germplasm Innovation & Utilization of Chinese Medicinal Materials in Southwest China, Yunnan Agricultural University, Kunming, Yunnan, China

[3]Yunnan Tobacco Company Yuxi Branch, Kunming, Yunnan, China

[4]Yunnan Tobacco Company Kunming Branch Songming Branch, Kunming, China

[5]Thai Nguyen University of Agriculture and Forestry, Thai Nguyen, Vietnam

## AUTHOR ORCIDs

Jili Chen ⓘ http://orcid.org/0009-0001-3344-9504
Jiabang Dong ⓘ http://orcid.org/0009-0007-3666-414X
Zhiwen Xiao ⓘ http://orcid.org/0009-0009-3106-7631
Xiaojun Zhao ⓘ http://orcid.org/0009-0004-7631-5241
Tao Liu ⓘ http://orcid.org/0000-0003-3585-4842

## AUTHOR CONTRIBUTIONS

Jili Chen, Formal analysis, Software, Writing – original draft, Writing – review and editing | Jiabang Dong, Resources, Software, Visualization, Writing – original draft, Writing – review and editing | Zhiwen Xiao, Investigation, Software, Visualization, Writing – original draft | Dong Jiang, Data curation, Investigation, Resources | Jialong Yu, Formal analysis, Software, validation | Lương Hùng Tiến, Software, Visualization | Hoàng Trung Tín, Investigation, Resources, Visualization | Xiaojun Zhao, Conceptualization, Data curation, Funding acquisition, Project administration, Writing – original draft | Tao Liu, Conceptualization, Funding acquisition, Methodology, Project administration, Supervision

## DATA AVAILABILITY

The whole genome of *B. velezensis* D83 has been uploaded to the NCBI database under accession number CP156684.1.

## ADDITIONAL FILES

The following material is available online.

### Supplemental Material

**Table S1 (Spectrum00300-25-s0001.xls).** Metabolites of strain D83.

### Open Peer Review

**PEER REVIEW HISTORY (review-history.pdf).** An accounting of the reviewer comments and feedback.

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
