## [Reviewer comments · Microbiology Spectrum]

Microbiology Spectrum

Genomics and metabolomics assisted functional characterization of *Bacillus velezensis* D83 as a biocontrol and plant growth promoting bacterium

Jili Chen, Jiabang Dong, Zhiwen Xiao, Dong Jiang, Jialong Yu, LƯƠNG HÙNG TIẾN, HOÀNG TRUNG TÍN, Xiaojun Zhao, and Tao Liu

Corresponding Author(s): Tao Liu, Yunnan Agricultural University

Review Timeline:

Submission Date:	January 28, 2025
Editorial Decision:	March 25, 2025
Revision Received:	June 6, 2025
Accepted:	June 11, 2025

Editor: Artem Rogovsky

Reviewer(s): Disclosure of reviewer identity is with reference to reviewer comments included in decision letter(s). The following individuals involved in review of your submission have agreed to reveal their identity: Changyi Zhang (Reviewer #2)

Transaction Report:

DOI: <https://doi.org/10.1128/spectrum.00300-25>

Re: Spectrum00300-25 (Genomics and metabolomics assisted functional characterization of *Bacillus velezensis* D83 as a biocontrol and plant growth promoting bacterium)

Dear Prof. Tao Liu:

Thank you for the privilege of reviewing your work. Below you will find my comments, instructions from the Spectrum editorial office, and the reviewer comments.

Revision Guidelines

Sincerely,
Artem Rogovsky
Editor
Microbiology Spectrum

Reviewer #2 (Comments for the Author):

The manuscript by Chen et al. describes the isolation of *Bacillus velezensis* D83, a strain with promising potential for biocontrol and plant growth promotion. The authors sequenced the D83 genome and conducted various bioinformatic analyses, including phylogenetics, comparative genomics, and secondary metabolite biosynthesis gene cluster analysis, aiming to identify genes or gene clusters involved in antimicrobial metabolite production. Additionally, they experimentally demonstrated that D83 exhibits

inhibitory activity against the plant fungal pathogens *Fusarium oxysporum* and *Phytophthora nicotianae*.

Overall, the study presents an interesting discovery, particularly regarding the potential of D83 as a biocontrol agent and a source of antimicrobial compounds. However, the manuscript requires significant improvements in data presentation, discussion, and writing quality. My specific comments are outlined below.

Major concerns:

1. This is lack of in-depth analysis of the genomic analysis of D83. Many sections present general findings without providing meaningful insights. A more detailed analysis and interpretation are needed.
2. The authors do not sufficiently highlight the significance and broader implications of their findings. The "Significance" section largely repeats information from the "Abstract", rather than providing a strong justification for the study's importance in the field.
3. While language quality should not be the primary criterion for evaluating research merit, the manuscript contains numerous incomplete and unclear sentences, which hinder readability and comprehension. Some examples are provided below.
4. There is lack of proper discussion in "Results and discussion part". Do closely related *Bacillus velezensis* strains also exhibit antimicrobial activity? Do these strains share the same gene clusters responsible for antimicrobial metabolite production? A more comprehensive discussion comparing D83 with related strains is necessary.

Specific comments:

1. The paper contains numerous incomplete sentences, which affect readability. Here are a few examples:
Line 172-173: "Use of DNA of acceptable purity, concentration and integrity for library creation. Detection of library fragment size with Agilent 2100 Bioanalyzer (Agilent technologies, USA). "

Lines 183-188, and line 243: incomplete sentences.

I believe there are many more such cases throughout the manuscript. A careful review and revision are necessary.

2. Line 153: Change "Evolutionary" to "Phylogenetic".
3. Line 163: Use "g" instead of "rpm" for centrifugation.
4. Line 189: Clarify what is meant by "CRISPR structure." Are the authors referring to CRISPR arrays or CRISPR-Cas systems? The description should be more precise.
5. Line 211: Change "Siderophore are low molecular weight Deferoxamine synthesized by microorganisms under iron-deficient conditions." to "Siderophores are low-molecular-weight deferoxamines synthesized by microorganisms under iron-deficient conditions."
6. Line 229: Specify whether the percentage is vol/vol (v/v) or wt/vol (w/v)?
7. Line 287: It is not clear how phylogenetic analysis was performed. Please clearly state the basis of the phylogenetic analysis- was it conducted using core genomes or SNPs across the whole genome?
8. Lines 299-313. The authors use several vague terms in this section, making data interpretation unclear. It is difficult to discern meaningful insights regarding the similarities and differences between the genomes. Additionally, the text in the figure is unreadable, and Figure 2 lacks proper legends to clarify its content.
9. More details should be provided regarding the "8 CRISPRs," including the number of repeat-spacer arrays and the specific types of CRISPR-Cas systems present.
10. Line 443: In Table 1, include "Genome coordinates" when describing genomic locations ("From" and "To"). Additionally, clarify that "similarity" is based on nucleotide sequence comparison.
11. In Figure 8, what is the P-value for the presented data?

Reviewer #3 (Comments for the Author):

The title of the paper is 'genomics and metabolomics assisted functional characterization of *Bacillus velezensis* D83 as a biocontrol and plant growth promoting bacterium' and discusses the role of this bacteria against two fungal pathogens, F

oxysporum and *P. nictitanae* .

The authors show experimental data that this strain D 83 inhibits both pathogens and helps with other processes such as nitrogen fixation, siderophore production, amylase production etc that help with plant growth. The authors did pot experiments, and lab experiments to show the characteristics of this isolate.

The strain shows promising results as an antifungal agent in plate assays and having nitrogenase, cellulase activity. Lines 461 and 462 discuss a bacteriophage which does not show a correlation with the indole acetic acid activity.

More figures should be shown showing the experiments of 3.5.1 and images of pot experiments should be added to the supplemental figures. I am not sure if fresh and dry mass data was recorded.

Using bioinformatics tools, specifically genome mining analysis, 102 genes were found to produce secondary metabolites that include antifungal metabolites. They also did broth analysis to check for the presence of these antifungal metabolites using LC/MS

The M/S has substantial data on genome annotation, comparative genome analysis of this strain. Strain D83 was found to belong to a monophyletic group and has members that have plant growth promoting ability and biocontrol ability.

The title of the paper is 'genomics and metabolomics assisted functional characterization of *Bacillus velezensis* D83 as a biocontrol and plant growth promoting bacterium' and discusses the role of this bacteria against two fungal pathogens, *F oxysporum* and *P. nicaiana*.

The authors show experimental data that this strain D 83 inhibits both pathogens and helps with other processes such as nitrogen fixation, siderophore production, amylase production etc that help with plant growth. The authors did pot experiments, and lab experiments to show the characteristics of this isolate.

The strain shows promising results as an antifungal agent in plate assays and having nitrogenase, cellulase activity. Lines 461 and 462 discuss a bacteriophage which does not show a correlation with the indole acetic acid activity.

More figures should be shown showing the experiments of 3.5.1 and images of pot experiments should be added to the supplemental figures. I am not sure if fresh and dry mass data was recorded.

Using bioinformatics tools, specifically genome mining analysis, 102 genes were found to produce secondary metabolites that include antifungal metabolites. They also did broth analysis to check for the presence of these antifungal metabolites using LC/MS

The M/S has substantial data on genome annotation, comparative genome analysis of this strain. Strain D83 was found to belong to a monophyletic group and has members that have plant growth promoting ability and biocontrol ability.

Comments to the editor:

The paper has a title 'genomics and metabolomics assisted functional characterization of *Bacillus velezensis* D83 as a biocontrol and plant growth promoting bacterium'

The abstract is centered on the experiments done to provide proof of its plant growth promoting ability.

However, when I read the paper, the results sections 3.2, 3.2, 3.3 and 3.4 are about genome annotation, gene characterization, pathways etc. This amounted to 7 pages of the M/S but the wet lab experiments were found in only one section 3.5 and amounted to 1.5 pages. Some figures were provided in the supplemental data. Since this is a PGPR strain as the authors claim, there should be more data/experiments substantiating that.

In my opinion, the paper needs to be restructured and some more experimental evidence should be provided such as plant growth (in photographs) and other data that compared this stain against others.

Dear Editor and Reviewers:

It was a great joy to get a letter from you. Thank you and the reviewers for your hard work on our manuscript (Manuscript ID: Spectrum00300-25R1), “Genomics and metabolomics assisted functional characterization of *Bacillus velezensis* D83 as a biocontrol and plant growth promoting bacterium”. Those comments are valuable and very helpful for revising and improving our paper. After careful revise, we have resubmitted on official website of revised manuscript in “Microbiology Spectrum”, which we hope to meet with approval. We submitted both a Revised Manuscript and a Clean Version of it, and the Revised Manuscript retains the changes. The main corrections and the responses to the comments are displayed below point by point.

If you have any queries, please don't hesitate to contact me at the address below.

Thank you and best regards.

Yours sincerely,

Tao Liu

E-mail: yantao618@126.com

Reviewer #2 (Comments for the Author):

The manuscript by Chen et al. describes the isolation of *Bacillus velezensis* D83, a strain with promising potential for biocontrol and plant growth promotion. The authors sequenced the D83 genome and conducted various bioinformatic analyses, including phylogenetics, comparative genomics, and secondary metabolite biosynthesis gene cluster analysis, aiming to identify genes or gene clusters involved in antimicrobial metabolite production. Additionally, they experimentally demonstrated that D83 exhibits inhibitory activity against the plant fungal pathogens *Fusarium oxysporum* and *Phytophthora nicotianae*.

Overall, the study presents an interesting discovery, particularly regarding the potential of D83 as a biocontrol agent and a source of antimicrobial compounds. However, the manuscript requires significant improvements in data presentation, discussion, and writing quality. My specific comments are outlined below.

Major concerns:

1. This is lack of in-depth analysis of the genomic analysis of D83. Many sections present general findings without providing meaningful insights. A more detailed analysis and interpretation are needed.

Response: Thank you very much for your professional review of our article. We agree with your views. In the revised manuscript, I added the following D83 genomic analyses: (i) In-depth analysis of growth-promoting genes in the D83 genome to complement the mechanisms by which D83 promotes the growth of maize seedlings and tobacco seedlings; (ii) differences in basic genomic features and clusters of genes encoding secondary metabolites were compared between D83 and other *Bacillus* species. (iii) the D83 genome was amplified by PCR with specific primers to determine whether the genes for synthesis of the lipopeptide antibiotic substances fengycin, iturin, surfactin were present on the D83 genome.

2. The authors do not sufficiently highlight the significance and broader implications of their findings. The "Significance" section largely repeats information from the "Abstract", rather than providing a strong justification for the study's importance in the field.

Response: Thank you very much for your professional review of our article. We agree with your views. In the significance section, I have reworked the presentation of this section to emphasize the significance and broader implications of the findings of this study. Specifically changed to read, “We have the following objectives and significance...for sustainable agriculture”. The specific changes are on page 3, lines 123-135.

3. While language quality should not be the primary criterion for evaluating research merit, the manuscript contains numerous incomplete and unclear sentences, which hinder readability and comprehension. Some examples are provided below.

Response: Thank you for your careful review and professional comments on this manuscript. We agree with your views. In the revised manuscript, although the

structure of the whole article was reworked, I also made changes regarding the linguistic expression of the whole article. For example, change “tRNA prediction of the D83 genome using tRNA prediction of the D83 genome using tRNAscan-SE (2.0.9)” to “The tRNA prediction of the D83 genome was carried out using tRNAscan-SE (2.0.9)”.

In addition, I checked the use of all punctuation marks as well as spaces, etc. to improve the quality of the manuscript.

4. There is lack of proper discussion in "Results and discussion part". Do closely related *Bacillus velezensis* strains also exhibit antimicrobial activity? Do these strains share the same gene clusters responsible for antimicrobial metabolite production? A more comprehensive discussion comparing D83 with related strains is necessary.

Response: Thank you for your careful review and professional comments on this manuscript. We agree with your views. On page 15 of the manuscript, lines 529-534, I compare strain D83 with the cluster of secondary metabolic synthesis genes of the closely related *Bacillus velezensis* strain. These strains were also analyzed and found to possess antimicrobial activity, sharing some of the gene clusters responsible for the production of antimicrobial metabolites, including surfactin, macrolactin H, bacillaene, fengycin, difficidin, bacillibactin, and bacilysin. However, there were some differences, in particular D83 had an additional cluster of genes encoding butirosin A/B compared to the four strains in the common comparison. *B. amyloliquefaciens* GKT04 had one more gene cluster encoding plantazolicin compared to the other strains. The specific table is on page 17, line 549. Table 4.

Specific comments:

1. The paper contains numerous incomplete sentences, which affect readability. Here are a few examples:

Line 172-173: "Use of DNA of acceptable purity, concentration and integrity for library creation. Detection of library fragment size with Agilent 2100 Bioanalyzer (Agilent technologies, USA). "

Response: Thank you for your careful review and professional comments on this manuscript. We agree with your views. In order to ensure the readability and completeness of the sentence, I have modified the above sentence by changing it to "For library creation, the DNA used should be of acceptable purity, concentration, and integrity. The fragment size of the library was detected using the Agilent 2100 Bioanalyzer (Agilent Technologies, USA)." The specific changes are on page 5, lines 230-232.

Lines 183-188, and line 243: incomplete sentences.

Response: Thank you for your careful review and professional comments on this manuscript. We agree with your views. In lines 183-188, I made changes to all the sentences. Specifically: "The tRNA prediction of the D83 genome was carried out...was predicted using MinCED (v0.4.2)." The revised sentence is on page 5, lines 248-254.

2. Line 153: Change "Evolutionary" to "Phylogenetic".

Response: Thank you very much for your professional review of our article. Thank you for your very careful discovery of errors in the manuscript. We concur with your perspectives. In line 153, I have changed "Evolutionary" to "Phylogenetic". The specific change is now on page 6, line 255.

3. Line 163: Use "g" instead of "rpm" for centrifugation.

Response: Thank you very much for your professional review of our article. We agree with your views. By converting the RPM on the centrifuge, on page 5, line 222 I changed 10000 rpm to 9391 g.

4. Line 189: Clarify what is meant by "CRISPR structure." Are the authors referring to CRISPR arrays or CRISPR-Cas systems? The description should be more precise.

Response: Thank you for your careful review and professional comments on this manuscript. We agree with your views. The full name of the CRISPR sequence is clustered regularly interspaced short palindromic repeat sequences. Described in this study are CRISPR arrays. The CRISPR predictions in this study are the sequences corresponding to the CRISPR structures in the CRISPR-Cas system. The system is that this sequence works in conjunction with proteins encoded by its neighboring genes to form the highly conserved CRISPR/Cas system in prokaryotes.

5. Line 211: Change "Siderophore are low molecular weight Deferoxamine synthesized by microorganisms under iron-deficient conditions." to "Siderophores are low-molecular-weight deferoxamines synthesized by microorganisms under iron-deficient conditions."

Response: Thank you for your careful review and professional comments on this manuscript. We agree with your views. Again, I am deeply sorry to you for the reading problems caused by the language. In line 211, I have changed "Siderophore are low molecular weight Deferoxamine synthesized by microorganisms under iron-deficient conditions." to "Siderophores are low-molecular-weight deferoxamines synthesized by microorganisms under iron-deficient conditions." Specific changes are on page 4, lines 174-175.

6. Line 229: Specify whether the percentage is vol/vol (v/v) or wt/vol (w/v)?

Response: Thank you for your careful review and professional comments on this manuscript. The percentage mentioned on line 229 actually refers to the area ratio. The inhibition rate was assessed by comparing the radial growth of fungal colonies between experimental and control plates. The antagonistic ability of D83 against phytopathogenic fungi was compared by comparing plates inoculated with the pathogenic fungi alone with plates inoculated with dual cultures of both the

pathogenic fungi and D83. The inhibitory rate was quantified by measuring the radial extension of pathogenic fungal colonies on both control and treated plates. Inhibition rates were calculated based on the size of the measured fungal radius. Inhibition rate (100%) = (radius of control colony - radius of experimental colony)/radius of control colony × 100%. In addition, in the newly supplemented experiment of volatile gas diffusion inhibition of pathogenic fungi, the inhibition rate was calculated according to the above method. **Now on page 3, lines 154-161.**

7. Line 287: It is not clear how phylogenetic analysis was performed. Please clearly state the basis of the phylogenetic analysis-was it conducted using core genomes or SNPs across the whole genome?

Response: Thank you for your careful review and professional comments on this manuscript. We agree with your views. In this study, the phylogenetic tree was generated using Prokka to extract annotation information from genomic sequences to generate GFF3 files. The core genome was compared using roary, and then a phylogenetic tree was constructed using FastTree 2.1 by performing 1,000 bootstrap passes via the Approximate Maximum Likelihood (AML) method. **Therefore, this study is a phylogenetic analysis using the core genome.**

8. Lines 299-313. The authors use several vague terms in this section, making data interpretation unclear. It is difficult to discern meaningful insights regarding the similarities and differences between the genomes. Additionally, the text in the figure is unreadable, and Figure 2 lacks proper legends to clarify its content.

Response: Thank you for your careful review and professional comments on this manuscript. We agree with your views. **In lines 299-313, I re-analyze this section with clarity. The specific change is: “To assess ... certain regions.” on page 14, lines 477-486. Additionally, for Figure 2, I have made a slight modification to the figure, on the right side of the figure, to add the name of the strain and also to increase the clarity of the figure, which is now modified as Figure 7c. The specific change is on page 13, line 444. In addition, to enhance the comprehension of the figure's content, I have added a legend in Fig. 7c. This legend supplements the information presented**

and clarifies the significance of the color - coded blocks. The specific changes are on page 13, lines 450-452.

9. More details should be provided regarding the "8 CRISPRs," including the number of repeat-spacer arrays and the specific types of CRISPR-Cas systems present.

Response: Thank you for your careful review and professional comments on this manuscript. We agree with your views. In subsequent analyses I used MinCED and the CRISPR Cas Finder to analyze the number of repeat-spacer arrays and the specific types of CRISPR-Cas systems. Unfortunately, however, using MinCED and the CRISPR Cas Finder, the number of repeat-spaced arrays and the specific type of CRISPR-Cas system for D83 were not predicted. Due to time constraints, so I have not added any information about this for the time being. After that I will continue to experiment with other software or methods for this part of the data analysis.

10. Line 443: In Table 1, include "Genome coordinates" when describing genomic locations ("From" and "To"). Additionally, clarify that "similarity" is based on nucleotide sequence comparison.

Response: Thank you for your careful review and professional comments on this manuscript. We agree with your views. In Table 1 (now Table 3), I described genomic locations with the addition of genomic coordinates, and I described the location of genes on chromosomes. The revised table is on page 16, line 542-543. In addition, on page 15, line 521, I illustrate that the comparison of secondary metabolite gene clusters is based on nucleotide sequence comparisons.

11. In Figure 8, what is the P-value for the presented data?

Response: Thank you for your careful review and professional comments on this manuscript. We agree with your views. In the data analysis section, I performed a one-way ANOVA in GraphPad Prism 10 software on the experimental data obtained, and the data are expressed as the mean \pm standard deviation (SD) of three independent biological replicates. Asterisks indicate significant differences. The p-values are different from the first submission of the manuscript because I added some of the experiments and photographs. But I have labeled the P-values for each treatment in the figure notes for Figures 4, 5, and 6. The specific changes are on page 10, lines 387-394; page 11, lines 396-403; and page 12, lines 427-433.

Fig. 4. Influence of *B. velezensis* D83 as soil inoculant on growth of maize in pot experiments. (a) and (b) Treated maize plants. D83: Maize seedlings treated with *B. velezensis* D83 suspension; CK: Maize seedlings treated with sterile water; A: Maize seedlings treated with commercially available microbial preparations. (c) Plant height (ns $p = 0.0849$; ns $p = 0.9969$). (d) Leaf length (***) $p = 0.0003$; ns $p = 0.9082$). (e) Root length (*) $p = 0.0388$; ns $p = 0.2758$). (f) Biomass (** $p = 0.0048$; ns $p = 0.3853$). (g) Fresh weight of underground part (***) $p = 0.0003$; ns $p = 0.8352$). (h) Fresh weight of aboveground part (*) $p = 0.0188$; ns $p = 0.3421$). Data represent mean \pm SD of three biological replicates. Asterisks indicate significant differences.

Fig. 5. Influence of *B. velezensis* D83 as soil inoculant on growth of tobacco in pot experiments. (a) and (b) Treated tobacco plants. D83: Tobacco seedlings treated with *B. velezensis* D83 suspension; CK: Tobacco seedlings treated with sterile water; A: Tobacco seedlings treated with commercially available microbial preparations. (c) Plant height (** $p = 0.0017$; * $p = 0.0278$). (d) Leaf length (** $p = 0.0033$; ns $p = 0.0278$). (e) Root length (**** $p < 0.0001$; *** $p = 0.0002$). (f) Biomass (**** $p < 0.0001$; ns $p = 0.6484$). (g) Fresh weight of underground part (*** $p = 0.0007$; ns $p = 0.5547$). (h) Fresh weight of aboveground part (*** $p = 0.0001$; ns $p = 0.07475$). Data represent mean \pm SD of three biological replicates. Asterisks indicate significant differences.

Fig. 6. (a) Effect of D83 treatment on CAT activity in maize plants (ns $p = 0.0849$; $*p = 0.0040$). (b) Effect of D83 treatment on MDA content in maize plants (ns $p > 0.05$). (c) Effect of D83 treatment on PPO activity in maize plants (ns $p > 0.05$). (d) Effect of D83 treatment on CAT activity in tobacco plants ($***p = 0.0002$; ns $p = 0.0573$). (e) Effect of D83 treatment on MDA content in tobacco plants (ns $p > 0.05$). (f) Effect of D83 treatment on PPO activity in tobacco plants ($**p = 0.0025$; $*p = 0.0335$). D83: Seedlings treated with *B. velezensis* D83 suspension; CK: Seedlings treated with sterile water; A: Seedlings treated with commercially available microbial preparations.

Reviewer #3 (Comments for the Author):

The title of the paper is ‘genomics and metabolomics assisted functional characterization of *Bacillus velezensis* D83 as a biocontrol and plant growth promoting

bacterium’ and discusses the role of this bacteria against two fungal pathogens, *F. oxysporum* and *P. nicotianae*.

The authors show experimental data that this strain D 83 inhibits both pathogens and

helps with other processes such as nitrogen fixation, siderophore production, amylase production etc that help with plant growth. The authors did pot experiments, and lab experiments to show the characteristics of this isolate.

1. The strain shows promising results as an antifungal agent in plate assays and having nitrogenase, cellulase activity. Lines 461 and 462 discuss a bacteriophage which does not show a correlation with the indole acetic acid activity.

Response: Thank you for your careful review and professional comments on this manuscript. We agree with your views. Again, I apologize for my mistake. In lines 461 and 462 I incorrectly mentioned phage, but what I was trying to convey is that in Figure S5h it was observed that strain D83 with the addition of the chromogenic solution had a light pink coloration, suggesting that it has the ability to produce indoleacetic acid. Now, in the manuscript, I've revised this section. The specific changes are in lines 341-344. The specific picture is on page 9, line 349.

2. More figures should be shown showing the experiments of 3.5.1 and images of pot experiments should be added to the supplemental figures. I am not sure if fresh and dry mass data was recorded.

Response: Thank you for your careful review and professional comments on this manuscript. We agree with your views. Again, I apologize for my mistake. In 3.5.1, I added images of all the experiments in the body of the text. The exact change is on page 11, line 395. In addition, in the newly revised manuscript, I have added pictures of tobacco potting experiments, as well as data on plant fresh weight and enzyme activity for stress tolerance. A set of experiments and pictures of growth promotion on corn seedlings were also added to illustrate the growth-promoting effects of D83 on plants. And added germination experiments with D83 on seedlings of maize and tobacco. Seed germination experiments and seedling promotion experiments, both of which I compared with commercially available microbial formulations. And these data and figures I have added to the manuscript. The specific changes are on page 9,

lines 362; page 10, lines 386; and page 11, lines 395.

Fig. 3. Effect of D83 on seed germination. (a) *In vitro* germination experiments with maize seeds. (b) Maize seed germination rate. (c) *In vitro* germination of tobacco seeds. (d) Tobacco seed germination rate. D83: Seeds treated with *B. velezensis* D83 suspension; CK: Seeds treated with sterile water; A: Seeds treated with commercially available microbial preparations.

Fig. 4. Influence of *B. velezensis* D83 as soil inoculant on growth of maize in pot experiments. (a) and (b) Treated maize plants. D83: Maize seedlings treated with *B. velezensis* D83 suspension; CK: Maize seedlings treated with sterile water; A: Maize seedlings treated with commercially available microbial preparations. (c) Plant height (ns $p = 0.0849$; ns $p = 0.9969$). (d) Leaf length (** $p = 0.0003$; ns $p = 0.9082$). (e) Root length (* $p = 0.0388$; ns $p = 0.2758$). (f)

Biomass (** p = 0.0048; ns p = 0.3853). (g) Fresh weight of underground part (** p = 0.0003; ns p = 0.8352). (h) Fresh weight of aboveground part (* p = 0.0188; ns p = 0.3421). Data represent mean \pm SD of three biological replicates. Asterisks indicate significant differences.

Fig. 5. Influence of *B. velezensis* D83 as soil inoculant on growth of tobacco in pot experiments. (a) and (b) Treated tobacco plants. D83: Tobacco seedlings treated with *B. velezensis* D83 suspension; CK: Tobacco seedlings treated with sterile water; A: Tobacco seedlings treated with commercially available microbial preparations. (c) Plant height (** p = 0.0017; * p = 0.0278). (d) Leaf length (** p = 0.0033; ns p = 0.0278). (e) Root length (**** p < 0.0001; *** p = 0.0002). (f) Biomass (**** p < 0.0001; ns p = 0.6484). (g) Fresh weight of underground part (** p = 0.0007; ns p = 0.5547). (h) Fresh weight of aboveground part (** p = 0.0001; ns p = 0.07475). Data represent mean \pm SD of three biological replicates. Asterisks indicate significant differences.

3. Using bioinformatics tools, specifically genome mining analysis, 102 genes were found to produce secondary metabolites that include antifungal metabolites. They also did broth analysis to check for the presence of these antifungal metabolites using LC/MS. The M/S has substantial data on genome annotation, comparative genome analysis of this strain. Strain D83 was found to belong to a monophyletic

group and has members that have plant growth promoting ability and biocontrol ability.

Response: Thank you for your careful review and professional comments on this manuscript. In this research, our research group isolated the culturable strain D83 from the rhizosphere soil of healthy plants in a continuous cropping site of *P. notoginseng*. We have the following objectives and significance through the present study: (i) To evaluate the inhibitory activity of D83 against eight strains of phytopathogenic fungi, including *S. rolfsii*, *F. oxysporum*, *P. parasitica*, by the production of volatile gases and by the dual culture method; (ii) evaluating of the probiotic potential of D83 on plates; (iii) Evaluate the ability of D83 to promote plant growth and seed germination and compare it with common commercially available microbial agents; (iv) Genome mining and metabolome analysis to investigate the mechanism of D83 for plant growth and biocontrol. This study provides a research basis for the application of D83 as a PGPR and biocontrol agent in agricultural production to provide applicable solutions for sustainable agriculture.

Re: Spectrum00300-25R1 (**Genomics and metabolomics assisted functional characterization of *Bacillus velezensis* D83 as a biocontrol and plant growth promoting bacterium**)

Dear Prof. Tao Liu:

Your manuscript has been accepted, and I am forwarding it to the ASM production staff for publication. Your paper will first be checked to make sure all elements meet the technical requirements. ASM staff will contact you if anything needs to be revised before copyediting and production can begin. Otherwise, you will be notified when your proofs are ready to be viewed.

Sincerely,
Artem Rogovsky
Editor
Microbiology Spectrum